# Cooling Effect of Urban Blue and Green Spaces: A Case Study of Changsha, China

**DOI:** 10.3390/ijerph20032613

**Published:** 2023-02-01

**Authors:** Xinyi Qiu, Sung-Ho Kil, Hyun-Kil Jo, Chan Park, Wonkyong Song, Yun Eui Choi

**Affiliations:** 1Department of Geographic Information Science, Central South University of Forestry and Technology, 498 Shaoshan Nan Street, Changsha 410004, China; 2Department of Ecological Landscape Architecture Design, Kangwon National University, Gangwon-Dae-Hak-Gil 1, Chuncheon 24341, Republic of Korea; 3Department of Landscape Architecture, University of Seoul, Seoulsilipdaero 163, Seoul 02504, Republic of Korea; 4Department of Landscape Architecture, Dankook University, 119 Dandae-ro, Dongnam-gu, Cheonan 31116, Republic of Korea

**Keywords:** blue–green space planning, remote sensing, surface fitting, threshold, urban thermal environment

## Abstract

The cooling effects of blue–green spaces on the urban heat island effect are complex and different. The purpose of this study is to simulate how the cooling effect of blue–green space changes with its size and shape. The cooling effects of 53 green patches and 28 water bodies in Changsha were extracted based on Landsat images. A surface fitting model was used to quantitatively reveal the relationship between the cooling effect of blue–green space and its size and shape. The results show that the cooling effects of blue–green spaces were enhanced with the increasing size, and then would become stable after a certain range (threshold). Certain thresholds were identified between the blue and green space areas (2.98 ha and 3.15 ha, respectively) and the cooling distance, and between the blue and green space areas (4.84 ha and 4.92 ha, respectively) and the cooling magnitude. In addition, the green space with an area of 9.08 ha and landscape shape index (LSI) of 2.97 could achieve a better cooling distance (413.46 m); and the blue space with an area of 29.4 ha and LSI of 1.75 could achieve a better cooling magnitude (5.17 °C). These findings provide useful guidelines for urban planning and improving urban livability in other regions with terrain and climate conditions similar to Changsha.

## 1. Introduction

In recent decades, global warming, rapid urbanization expansion, an increase in population, and the loss of blue–green spaces have caused a continuous increase in land surface temperature (LST), raising a series of urban thermal environmental issues [1,2]. The deterioration of the urban thermal environment, such as the urban heat island effect (UHI), poses serious threats to the ecological environment as well as the health of residents [3,4,5]. For instance, approximately 70,000 additional heat-related deaths occurred in Europe in 2003 [6]. Alleviation of urban thermal environment issues by scientific strategies has become a necessity.

The UHI effect refers to the phenomenon in which higher LSTs occur in urban areas than in surrounding rural areas [7,8]. Urban blue–green spaces play a significant role in mitigating the urban thermal environment [9,10,11]. Blue–green spaces are defined as land covered by water or vegetation, including rivers, lakes, forests, grasslands, and croplands [12]. Generally, the temperatures of blue–green spaces are lower than those of built-up areas considering the different thermodynamic properties, which is known as the cooling island effect [13,14]. Previous studies have revealed that the cooling island effect of blue–green spaces can effectively reduce the UHI effect and heat stress, thus providing citizens with comfortable outdoor environments [15,16,17]. Over the past years, these studies have mainly concentrated on the spatial law, mechanism, and application of the urban cooling island effect. To better utilize the cooling effect of blue–green spaces, more scholars have further investigated the quantitative relationships between blue–green space characteristics and LST recently. The area and shape of blue–green spaces are commonly recognized as the determining factors influencing their cooling effect [18,19]. Blue–green spaces with larger areas usually have a better cooling effect [20], while several studies reported that green space patches with complex shapes have a better cooling effect than patches with simple shapes [21,22,23]. However, current quantitative data are still inadequate considering the differences in topography and climate for each region. Moreover, existing research has primarily focused on the relationship between a single variable and the LST, especially the size or shape of the blue–green space and the LST, and seldom examines the correlation between the two variables. The combination of the area and shape of blue–green spaces for optimal cooling effects is still unclear. Determining how to maximize the cooling effect of blue–green spaces in the limited spaces of urban areas with concentrated populations and high land utilization degrees provides important practical significance for the living environments of urban residents and the urban ecological level.

As one of the ten hottest cities in China, the urban heat island effect in Changsha is especially serious in August [24]. The purpose of this study is to explore how to achieve a better cooling effect of the blue–green spaces with different sizes and shapes based on Landsat images. The results will provide useful guidelines for the future urban planning of other countries and regions with terrain and climate conditions similar to those of Changsha.

## 2. Materials and Methods

### 2.1. Study Area

Changsha, as the capital of Hunan Province, is located in the midcentral region of China (Figure 1). The location of the city is in the lower reaches of Dongting Lake. Changsha has been called ‘a city of mountain and water’ for its abundant surface water resources, and the city is located in a basin with hills surrounding the urban zones. The city’s average air temperature in the hottest month is over 29.4 °C, with rainfall of 139.75 mm and an annual average air temperature of approximately 17.2 °C [25]. Up to 2016, Changsha developed rapidly, with an urbanization rate of 77.59% [26]. The city’s population increased to more than 10.04 million in 2020, approximately 2.2 times that in the 1980s (4.5 million), as reported by the Seventh National Census of Hunan Province. Combined with the local topography, the climate characteristics of hot summers, rapid urbanization, increased population, and the loss of blue–green spaces have together led to a continuous increase in the LSTs of Changsha, thus compounding these serious thermal environment problems and threatening residents’ health.

### 2.2. Landsat Image Collection

For Changsha, the most significant urban heat island effect occurs in August [27], i.e., the month with the highest temperature and the lowest rainfall frequency, since it is under the control of the subtropical anticyclone during this period. Thus, only the cooling effect of the blue–green space in August (the warmest season) was considered in this study. Limited by the temporal resolution of Landsat (16 days) and the effect of clouds, only 3 dated images (without clouds) were available in August (1998.08.23, 2009.08.21, 2019.08.17) from the 1990s to 2021, i.e., Landsat TM5 images dated 23 August 1998 and 21 August 2009 and Landsat 8 OLT/TIRS images dated 17 August 2019, with two scenes (path 123, row 40; path 123, row 4). The interval between two adjacent images is about ten years, and these 3 images exactly reflect the different states of Changsha’s urbanization process. All used Landsat images were collected during the daytime (10:35, 10:46, and 10:56 a.m. in local time, respectively) and obtained from the US Geological Survey (USGS) website. The method for LST retrieval and the data of blue–green spaces were presented in a previous work [28].

### 2.3. Definition of the Cooling Island Effect of Blue–Green Space

Considering the spatial resolution of the thermal infrared data are 120 m for Landsat TM5 and 100 m for Landsat 8, the blue–green spaces lower than 1.44 ha in 1998 and 2009 and lower than 1 ha in 2019 were removed to reduce the effect of mixed pixels. The spatial distribution of selected blue–green spaces is shown in Figure 2a. To reduce the cross-cooling effect between various cooling islands, a single blue–green space with no other green space or water body within 200 m was adopted as the research object (Figure 2b). During the research period, a total of 53 green spaces and 28 blue spaces were selected: 14 green spaces and 10 blue spaces were selected in 1998, 17 green spaces and 9 blue spaces were selected in 2009, and 22 green spaces and 9 blue spaces were selected in 2019.

To evaluate the cooling effect of blue–green space patches, eight straight lines were defined along the edges of every selected patch with a fixed length (1500 m) in eight sampling directions (Figure 2d). The original spatial resolution of the thermal infrared data of is 120 m for Landsat TM and 100 m for Landsat TIRS. However, the thermal infrared data were resampled to 30 m by the data producer by interpolation method before they were released on the website (USGS). Namely, the spatial resolution of the thermal infrared data used in this study is 30 m; thus, the sample distance is defined as 30 m to analyse the cooling effect in different directions. The LST was then identified at distances every 30 m along these straight lines, and the mean LST of the eight lines was calculated to describe the LST at a given corresponding distance. Two indicators, including the cooling distance and cooling magnitude, were selected to evaluate the cooling island effect of the blue–green space on the surrounding environment [29,30], which could be acquired by the first turning point on the LST curve (Figure 2c). More specifically, the cooling distance was the distance between the edge of a single cooling island and the first turning point on the LST curve, and the cooling magnitude was the temperature difference between a single cooling island and the first LST turning point.

### 2.4. Statistical Analysis

Area and shape have been shown to be the dominant factors affecting the cooling effect of blue–green spaces [31]. The landscape shape index (LSI), as an indicator of the complexity of the patch shape, is generally applied to investigate the influence of the shape of blue–green spaces on their cooling effect. Therefore, herein, we chose the area and landscape shape index (LSI) to investigate the influence of the characteristics of blue–green spaces on their cooling effect.

The LSI is applied to reflect the shape of the patches, which can be calculated as:(1)$$LSI=\frac{D}{2\sqrt{\pi \times A}}$$
where *D* is the perimeter of a single blue–green space patch and *A* is the area of a single blue–green space patch. Note that *D* and *A* of single blue–green space extracted from remote sensing images are approximate values. Larger *LSI* values indicate a more complicated shape, where the shape is a circle for *LSI* = 1 and a square for *LSI* = 1.13 [32].

Previous studies indicated that the cooling effect has a positive relationship with patch size. However, with a further increasing area, the increase in the cooling effect shows a trend of first decelerating and then stabilizing within a certain range [33]. To reveal this characteristic, logarithmic regression analysis was performed herein to provide an approximate relationship between the cooling distance/magnitude and blue–green space area; a threshold value was identified at the inflection point corresponding to the decelerated change in the slope of the fitting logarithmic curve. Note that the blue–green space with this threshold area could theoretically achieve the maximum cooling efficiency [34].

Surface fitting is a common method to obtain relevant and general information from the system under examination, and this method is widely used in engineering sciences [35,36,37]. To systematically describe the relationship between the size, shape, and cooling effect, the nonlinear surface fitting method is introduced to establish a quantitative model to obtain a better cooling effect with a given area and LSI. The equation can be expressed as:(2)$$f\left(x,y\right)=a_{0}+a_{1}x_{1}^{2}+a_{2}x_{2}^{2}+a_{3}x_{1}x_{2}$$
where *f*(*x*,*y*) is the cooling effect, for example, the cooling distance or cooling magnitude; *x*_1_ is the area; and *x*_2_ is the LSI. All processes were completed by Origin Pro 2018 [38].

## 3. Results and Discussion

The cooling island effect of each blue–green space shows the trend of increasing LSTs as the distance of the cooling island increases; however, after reaching a certain distance, the change in LST begins to fluctuate or stabilize, showing that the cooling island effect has a certain range of influence.

### 3.1. Cooling Effect of Green Spaces

Among the 53 analysed green spaces (Table A1), the maximum and minimum areas of the extracted green spaces were 1.02 and 14.64 ha, respectively; the LSIs ranged from 1.04~2.97; and the average LST was 30.34 °C. The cooling distances of green spaces ranged from 30~340 m, with an average of 169.25 m; and the cooling magnitudes of green spaces ranged from 0.7~3.54 °C, with an average of 1.85 °C in 53 green spaces.

The cooling distances of the green spaces were positively correlated with the green space areas and LSIs, indicating that a larger area and more complex shape led to longer cooling distances. The results show that a significant threshold was identified in the green space area with regard to the cooling effect; however, in the LSI results, no noticeable threshold relationship was found with the cooling effect.

The green space area was found to be the dominant factor affecting the cooling distance and cooling magnitude (Figure 3). The results show that the cooling distances of green spaces increase when the area is smaller than 3.15 ha. The rate of increase slows down significantly when the area exceeds 3.15 ha. This indicates a threshold in the green space area with regard to the cooling distance; a green space with an area of 3.15 ha could influence a cooling distance of 190.92 m with optimal efficiency. Similarly, a threshold corresponding to the green space area of 4.92 ha could reach a cooling magnitude of 2.29 °C with optimal efficiency.

The changes in the areas and LSIs of green spaces significantly influenced the cooling distance; the green space with an area of 9.08 ha and LSI of 2.97 could achieve a cooling distance of 413.46 m (Figure 4a). Among all selected samples (53 green spaces), the largest cooling distance was 340 m, corresponding to a green space area of 9.33 ha and an LSI of 1.57 (Figure 4b), and the smallest cooling distance was 30 m, corresponding to a green space area of 1.03 ha and an LSI of 1.06 (Figure 4d). In addition, the cooling distance of a single random green space with an area of 5.97 ha and an LSI of 1.5 was 240 m (Figure 4c). Therefore, to increase the cooling distance, it is recommended to enlarge the size of the green space and increase the complexity of the edge shape of the green space.

The changes in the areas and LSIs of green spaces significantly influenced the cooling magnitude (Figure 5a); the green space with an area of 14.64 ha and LSI of 1.88 achieved a cooling magnitude of 3.98 °C. Among all selected samples (53 green spaces), the largest cooling magnitude was 3.54 °C, corresponding to a green space with an area of 9.33 ha and an LSI of 1.57 (Figure 5b), while the smallest cooling magnitude was 0.7 °C, corresponding to a green space with an area of 1.29 ha and an LSI of 1.1 (Figure 5d). In addition, the cooling magnitude of a single random green space with an area of 3.56 ha and an LSI of 1.52 was 2.06 °C (Figure 5c). Therefore, to increase the cooling magnitude, it is recommended that green spaces with a certain size and highly complex shape be designed to expand the area of contact with the surroundings.

### 3.2. Cooling Effect of Blue Spaces

Among the 28 analysed blue spaces (Table A2), the areas of extracted blue spaces ranged from 1.01~29.99 ha, while the LSIs ranged from 1.04~1.75, and the average LST was 29.28 °C. The cooling distances of the blue spaces ranged from 63.75~370 m (average of 175.58 m) and the cooling magnitudes of the blue spaces ranged from 0.73~5.04 °C (average of 2.0 °C). These results show that the cooling effect of blue spaces is better than that of green spaces. A relatively small blue space area can achieve the same cooling effect as a large green space area, indicating that blue spaces have superior cooling efficiencies than green spaces.

Similar to the findings obtained for green spaces, the cooling distances of the blue spaces were positively correlated with their area and LSI, indicating that a larger area and more complex shape led to longer cooling distances. A significant threshold was identified in the area of blue spaces with regard to the cooling effect; however, no obvious threshold relationship was found between the LSI and the cooling effect.

The area of a blue space is the dominant factor affecting its cooling distance and cooling magnitude (Figure 6). A threshold corresponding to the blue space area of 2.98 ha could reach a cooling distance of 192.78 m with optimal efficiency, whereas a threshold corresponding to the blue space area of 4.84 ha could reach a cooling magnitude of 2.44 °C with optimal efficiency. Therefore, blue spaces can achieve a certain cooling effect with a relatively small size, indicating that the cooling distance efficiency of blue spaces is better than that of green spaces.

The changes in the areas and LSIs of blue spaces significantly influenced the cooling distance (Figure 7a); the blue space with an area of 24.08 ha and LSI of 1.75 provided cooling at a distance of 391.37 m (Figure 7a). Among all selected samples (28 blue spaces), the largest cooling distance was 370 m, corresponding to a blue space area of 29.99 ha and an LSI of 1.52 (Figure 7b), while the smallest cooling distance, 63.75 m, corresponded to a blue space with an area of 1.03 ha and an LSI of 1.06 (Figure 7d). In addition, the cooling distance of a single random blue space with an area of 5.05 ha and an LSI of 1.06 was 202.5 m (Figure 7c). The blue spaces with small sizes and simple shapes had the shortest cooling distances.

The changes in the blue space area and LSI significantly influenced the cooling magnitudes of blue spaces, and this model indicated a higher explanatory rate than that obtained for green spaces (Figure 8a). This was because blue spaces are generally distributed in the suburbs and surrounded by green spaces; thus, the land use type differences in the regions surrounding blue spaces are smaller than those of the regions surrounding green spaces, leading to a smaller LST range in the areas surrounding blue spaces than green spaces. Therefore, the cooling effect of blue spaces is mainly determined by their size and shape. The results show that a blue space with an area of 29.4 ha and an LSI of 1.75 could achieve a cooling magnitude of 5.17 °C.

Among all selected samples (28 blue spaces), the largest cooling magnitude was 5.04 °C with an area of 29.99 ha and an LSI of 1.52 (Figure 8b), while the smallest cooling magnitude was 0.73 °C with an area of 1.01 ha and an LSI of 1.06 (Figure 8d). In addition, the cooling magnitude of a single random blue space with an area of 10.96 ha and an LSI of 1.58 was 3.04 °C (Figure 8c). A larger blue space size achieves a larger cooling magnitude; however, the shape of the blue space does not need to be complex. Overall, the cooling magnitudes of the blue spaces were larger than those of the green spaces. This can be explained by the lower LSTs in the water body regions resulting from the higher heat capacity of water and its slow temperature rise process, causing larger LST differences between the water bodies and surrounding areas. The cooling magnitudes of some small green spaces were better than those of blue spaces, likely due to the locations of the green spaces; small green spaces are commonly located in urban centres. In this situation, the LST differences between the green spaces and the surrounding areas may also be greater.

### 3.3. Discussion

This research reveals that the average cooling distance and cooling magnitude of green spaces are 163.33 m and 1.8 °C, respectively, while the average cooling distance and cooling magnitude of blue spaces are 175.58 m and 2 °C, respectively. These findings differ from the results of previous studies; for instance, Du et al. [20] found that a blue–green space with an area of 1.12 ha had the lowest cooling distance (90 m) and cooling magnitude (0.78 °C), while that of 129.46 ha had the highest cooling distance (1610 m) and cooling magnitude (9.35 °C). Sun and Chen [39] indicated that 91% of the cooling effect of water bodies was exhibited within a range of 600 m. Park found that 300 m^2^ of green space could reduce the temperature by 1 °C, while 650 m^2^ of larger parks could form a temperature reduction by up to 2 °C [40]. Zeng et al. [41] revealed that 2 m^2^ of water could cool the environment by 0.6 °C. These differences could be explained by four aspects: (1) differences in the selected blue–green space samples representing different study regions with various topographic and geomorphologic conditions, including the depth and volume of blue spaces and NDVI in green spaces, (2) differences in urban climate and Landsat images date, (3) differences in the urbanization levels of the studied cities, and (4) differences in the impact of different remote sensing satellite sensors.

The results obtained herein reflect different cooling effects of blue and green spaces; blue spaces had a superior cooling effect than green spaces. This finding is consistent with previous studies. For example, Lin et al. [5,42] and Yu et al. [43] pointed out that water bodies have higher cooling efficiency than green spaces. This may be explained by the fact that green and blue spaces provide cooling effects through different mechanisms [17,44,45]. More specifically, blue spaces (water bodies) have larger thermal inertia and heat capacities than green spaces. Lower heat conduction and heat emissivity effectively reduce the sensible heat exchange capacity and further change the energy transmission [9,31,46,47]. In addition, water with a significant transpiration effect shows more advantages in improving the local microclimate and alleviating the thermal environment problem than green spaces [19,48].

In this research, we determined that the efficient threshold sizes were 3.15 ha (for the green space cooling distance), 4.92 ha (for the green space cooling magnitude), 2.98 ha (for the blue space cooling distance), and 4.84 ha (for the blue space cooling magnitude). These threshold size values represent the change points of the cooling efficiencies of blue–green spaces; past these change points, the cooling effect per unit area of blue–green spaces does not increase significantly [22,43,49]. For example, if a city’s estimated threshold size value (of green spaces) is 3 ha, it means that a green space area of 3 ha could achieve the optimal cooling efficiency; therefore, it is not necessary to build 10 ha of green space to reduce LSTs. Such results would be helpful for planners to use the smallest possible blue–green spaces to obtain a superior cooling effect. In addition, previous studies have also confirmed the existence of threshold size values. However, the past-reported threshold size values were different among various regions [34]; for example, area thresholds of 14 ha (Chongqing), 3 ha (Taipei), 5.6 ha (Leipzig), 1 ha (Hanoi), and 0.69 ha (Copenhagen) have all been reported. Note that the threshold size values are not only related to the differences in the definition of the cooling effect but also may be influenced by the climatic conditions [50], location, and latitude of the study area and could be altered based on research data collected at different resolutions or different scales [34].

## 4. Conclusions

How the cooling effect of blue–green spaces changes with their sizes and shapes in Changsha (China) was quantitatively simulated based on Landsat images in this study area. The average cooling distances of green and blue spaces were found to be 163.33 m and 175.58 m, respectively, while the average cooling magnitudes of green and blue spaces were 1.8 °C and 2 °C, respectively. In addition, certain thresholds were found to influence the cooling effects of blue–green spaces, where the cooling effect would slow down after the blue–green space area reached these identified thresholds. Meanwhile, a more efficient cooling effect was obtained at these thresholds. The thresholds were 3.15 ha for the green space cooling distance, 2.98 ha for the blue space cooling distance, 4.92 ha for the green space cooling magnitude, and 4.84 ha for the blue space cooling magnitude; these values are applicable in urban planning projects to achieve more efficient possible blue–green spaces. Furthermore, it was found that 9.08 ha of green space with an LSI of 2.97 could achieve a cooling distance of 413.46 m, while 24.08 ha of blue space with an LSI of 1.75 could achieve a cooling distance of 391.37 m. Additionally, 14.64 ha of green space with an LSI of 1.88 could achieve a cooling magnitude of 3.98 °C, while 24.08 ha of blue space with an LSI of 1.75 could achieve a cooling magnitude of 5.17 °C. The findings provide useful guidelines for the future urban planning of Changsha and other cities facing similar issues.

However, there are some limitations that need to be noted. The internal factors of the blue–green space itself, such as the density, type, proportion of different types, and height of green spaces, and the depth, volume, and water amount of blue spaces; and the external factors, such as landscape indicators, wind speed, and wind direction, were not considered in the study. Thus, there are still many works needed to be done in the future if more kinds of remote sensing data and meteorological data are available. Moreover, since the excessive cooling may provide more issues, the cooling effect of blue–green spaces in different seasons, especially in winter, needs to be studied in the future. In addition, this research was based on the subtropical monsoon climate of Changsha (China). It could be further carried out in regions with different climatic characteristics to find differences and commonalities in the cooling effects of blue–green spaces under different climatic characteristics. Therefore, the study would provide theoretical guidance and practical suggestions for planning blue–green spaces and for formulating policies aiming to alleviate urban thermal environment issues in other regions with diverse climatic characteristics.

## Figures and Tables

**Figure 1 ijerph-20-02613-f001:**
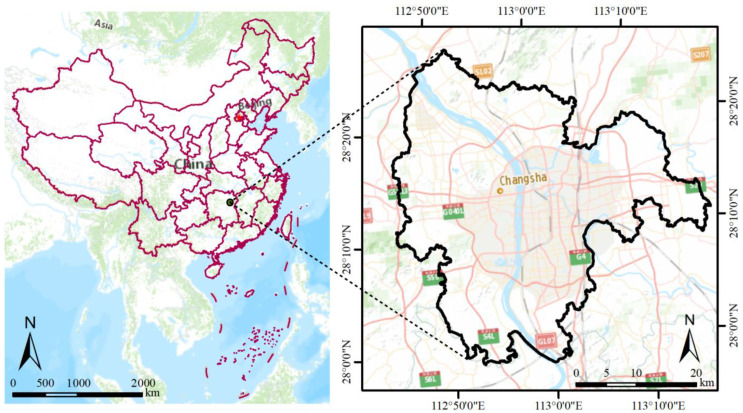
Study area.

**Figure 2 ijerph-20-02613-f002:**
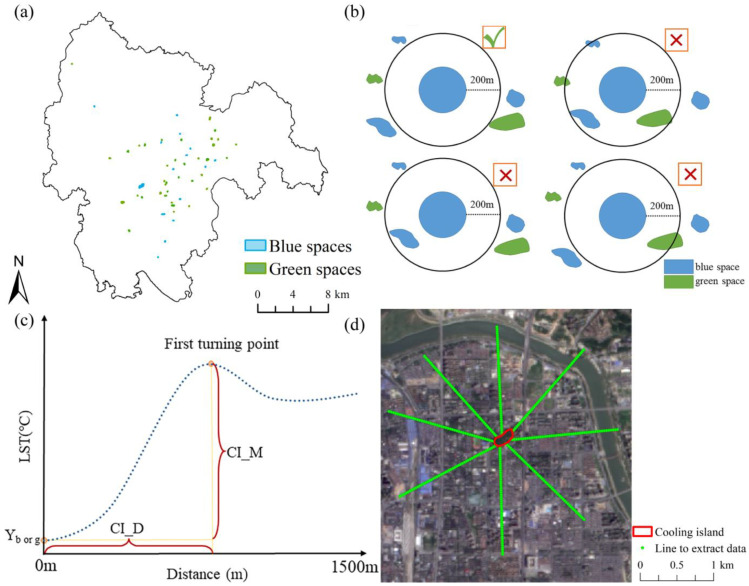
The cooling island effect of blue–green spaces: (**a**) distribution of green–blue spaces; (**b**) selection of blue–green spaces; (**c**) schematic diagram of the cooling island effect of blue–green space, where Y is the mean LST in a single blue or green space, CI_D is the cooling distance (m), and CI_M is the cooling magnitude (°C); and (**d**) eight sampling directions.

**Figure 3 ijerph-20-02613-f003:**
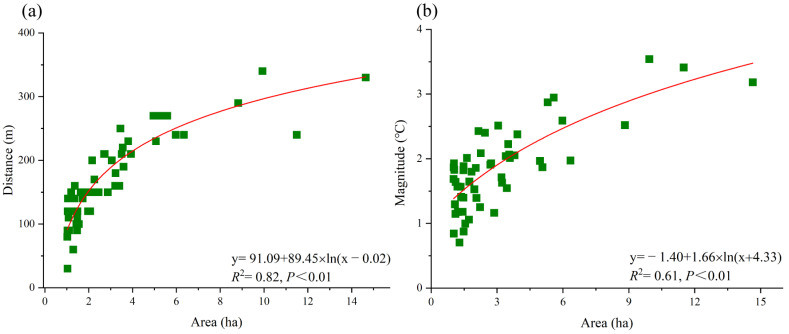
Relationship between the green space area and cooling distance/magnitude. (**a**) area VS cooling distance; (**b**) area VS cooling magnitude.

**Figure 4 ijerph-20-02613-f004:**
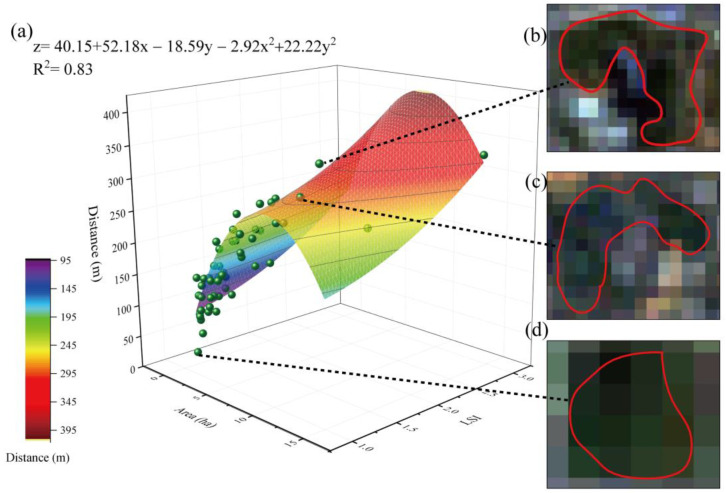
Relationships among green space area, LSI, and cooling distance. (**a**) Function description; (**b**) sample 1, a green space with an area of 9.33 ha and LSI of 1.57; (**c**) sample 2, a green space with an area of 5.97 ha and LSI of 1.50; and (**d**) sample 3, a green space with an area of 1.03 ha and LSI of 1.06.

**Figure 5 ijerph-20-02613-f005:**
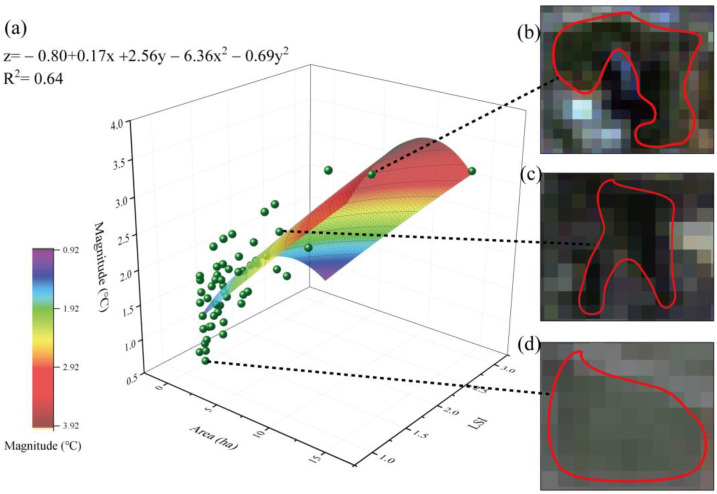
Relationships among the green space area, LSI, and cooling magnitude. (**a**) Function description; (**b**) sample 1, a green space with an area of 9.93 ha and LSI of 1.57; (**c**) sample 2, a green space with an area of 3.56 ha and LSI of 1.52; and (**d**) sample 3, a green space with an area of 1.29 ha and LSI of 1.10.

**Figure 6 ijerph-20-02613-f006:**
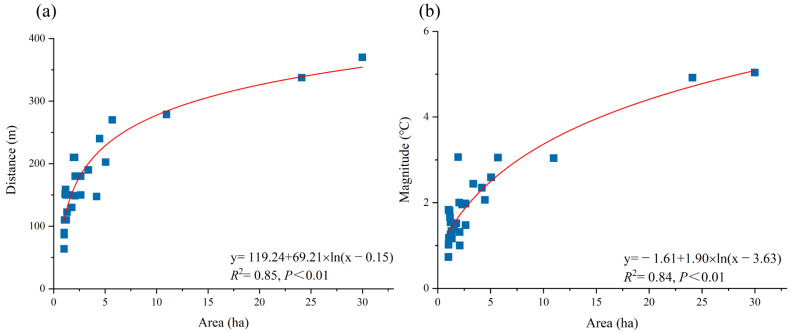
Relationship between blue space area and cooling distance/magnitude. (**a**) area VS cooling distance; (**b**) area VS cooling magnitude.

**Figure 7 ijerph-20-02613-f007:**
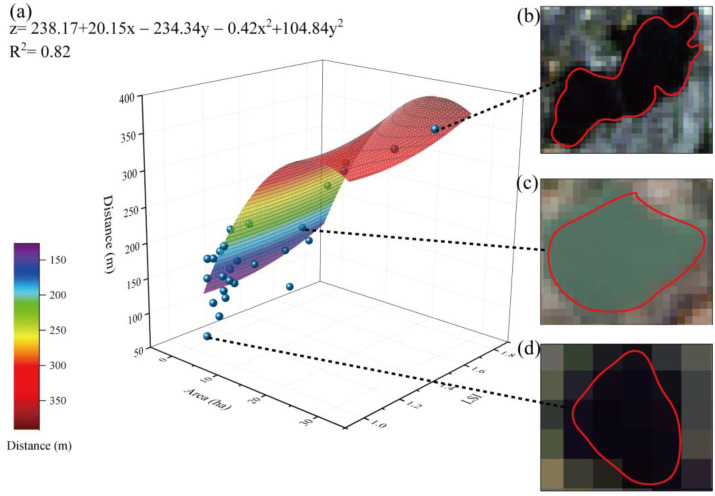
Relationships among the blue space area, LSI, and cooling distance. (**a**) Function description; (**b**) sample 1, a blue space with an area of 29.99 ha and LSI of 1.52; (**c**) sample 2, a blue space with an area of 5.05 ha and LSI of 1.06; and (**d**) sample 3, a blue space with an area of 1.01 ha and LSI of 1.06.

**Figure 8 ijerph-20-02613-f008:**
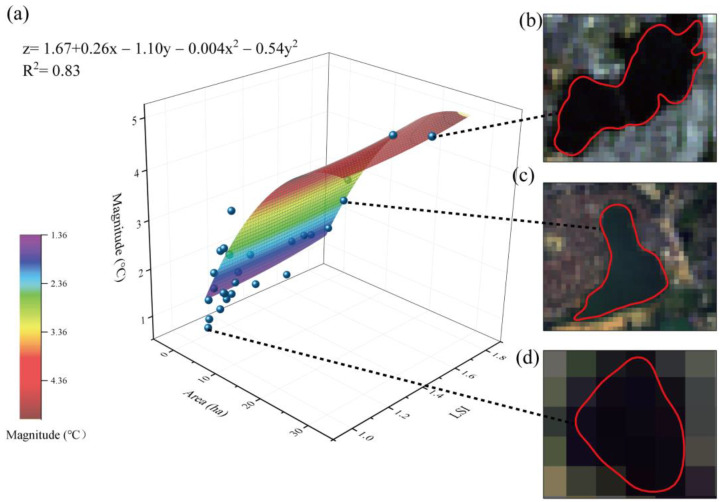
Relationships among the blue space area, LSI, and cooling magnitude. (**a**) Function description; (**b**) sample 1, a blue space with an area of 29.99 and LSI of 1.52; (**c**) sample 2, a blue space with an area of 10.96 ha and LSI of 1.58; and (**d**) sample 3, a blue space with an area of 1.01 ha and LSI of 1.06.

## Data Availability

No data were used for the research described in the article. Data will be made available on request.

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
