# Peer review of "Cooling Effect of Urban Blue and Green Spaces: A Case Study of Changsha, China"

_ijerph, 2023, doi:10.3390/ijerph20032613_

Round 1

Reviewer 1 Report

This study is significant in that it was confirmed the distance and magnitude of the cooling effect in urban areas by blue and green space using satellite imagery. The research results are also in a range of values that are sufficiently accepted. However, the writing style and structure of the paper seems to need revision. It is especially necessary to modify paragraph structure and expression skills within each session.

1. In abstract, background, research purpose, methods, results, and conclusions should be condensed and described, but in this paper just only the methods and results are described in detail, so they cannot serve as an abstract. Please rewrite the abstract.

2. In the introduction, there is a part that overlaps with the study area. Redundant content should be deleted, and only content related to the important reason for selecting the study area should be written.

3. For tables 1 and 2, I recommend arranging according to the green or blue space area. Because the site number is a value that is not very important to readers. It is also recommended that these tables be moved to an appendix.

4.  It would be helpful for readers to understand this study if the results of blue and green space were visually differentiated by different colors.

5. It seems that the conclusion is not organized well. The content is fine, but one sentence to one paragraph is a writing style to avoid. Correct the paragraph and sentence structure in this session.

Author Response

Thanks for your  suggestion. We have carefully checked the contents and revised the manuscript. All revisions are marked up using the “Track Changes” function. Point-by-point responses to all comments and questions are shown in the attachment.

Reviewer 2 Report

Dear Authors, 

thanks for the impressive work you have done and for having me the opportunity to review it.

I think that this paper could benefits of an improved description of the main parameters used (such as Aera and Perimeters definition, Landsat GSD used, increase of the number of the iages used, consideration on seasonal effect) .

It is correct that you used images acquired in the same month, but i suggest to extend this approach also in winter time to estimate the minimum threshold during cooler time.

I do really appreciate your work and my intent is only to suggest you how to improve it also to reduce the uncertanties 

Author Response

(The authors gave the same response as above.)

Reviewer 3 Report

There have been a lot of studies on shape and cold island effect, and this paper does not propose more innovations.

A large number of studies have shown that there are many factors affecting the cold island effect. The three-dimensional green amount of the park,  and the volume ratio, the building density, the proportion of the impervious surface area, the proportion of the green coverage area, the proportion of the trees, shrubs and grasses, the wind speed of the surrounding environment, have an effect on the cold island effect. The paper judged the impact of the cold island effect based solely on the landscape shape index of parks or water bodies. I think the results are not reliable and therefore not recommended for publication.

Author Response

Thanks for your comments. Yes, there have been a lot of studies on shape and cold island effect. However, as you said, there are still many questions in this field.

First, many studies on cold island effect focus on the northern cities, studies for cities in subtropical region is few. The cold island effect in different climate zones are different. The climate in Changsha (one of the ten hottest city in China) is typical, which can represent the similar city located in the southeast hills of China. For Changsha, the highest temperature occurs in August, but the largest rainfall occurs in June, the state with the highest temperature and the lowest rainfall (affected by the strong subtropical high pressure cyclone) in August expands the urban heat island effect in this period. Thus, findings of the studies on urban heat island effect and the cooling effect of the blue-green space in Changsha will benefit other cities with similar climate characteristics.

Second, there are few studies quantitatively simulate how the cooling effect change with distance, area and shape. Thus, the finding of this study will provide more accurate information on how shape and area of the blue-green space affects its cooling effect.

Third, yes, as you said, the three-dimensional green amount of the park, and the volume ratio, the building density, the proportion of the impervious surface area, the proportion of the green coverage area, the proportion of the trees, shrubs and grasses, the wind speed of the surrounding environment, have an effect on the cold island effect. However, the obtain of some data is difficult, e.g., three- dimensional green amount of the park, the volume ratio, and we would like to do more works according to your nice suggestion in the future.

Reviewer 4 Report

The subject is interesting and, as presented in the article, already analyzed by other studies as well. In my opinion the research is relevant and clearly presented. Rephrasing could improve the article in some parts that have to long sentences but is not mandatory.   

The interesting thing is that some findings are enforcing previous findings but some differences emerge, due to different climatic conditions, topography and level of urbanization, as stated by authors. I believe that the study could be improved in the future with an analysis that measure the body of water of bule areas, as well as the depth of them. For the green areas the density, height and even species of trees could be relevant in a way and give more answers and control for the results.  

Regarding the reviewed article, as I've said the subject is very interesting but is not the first time that the green and blue areas and their relation with UHI is analysed . The results of the research are coherent and the information are recent. The study is somehow superficial because analyses only general information available and does not relate those with green area density, species percentage types of trees and so on, information that could present different elements that influence the results. For the blue area the study could be improved with measurements of the depth or volume of water related to the surface because a 30 cm deep lake its getting worm faster than a 20 m deep lake and so on. The study could be improved with those kind of comparisons that would generate more substance for discussions and concision. Also the effect of UHI should be analysed for winter as well because the excessive cooling could provide more issues. 

Regarding the improvements that are needed for the structure of the article there are some minor issues. First, the introduction part could be rephrased and compacted and some elements that are repeated in the next parts could be excluded.  For example rows 51-86 need a revision and some parts can be used in Methods part or as references in the discussion part to relate tot the study findings. 

A general look over the whole article is needed because are more parts that could be rephrased. Some long phrases could be shortened and very short entrances should be avoided.

Author Response

(The authors gave the same response as above.)

Round 2

Reviewer 3 Report

For now, the paper is acceptable